evolution/ecology/behaviour

early life effects, reproductive development, female competition, maternal effects, sperm competition

**Author for correspondence:**
Liane Hobson
e-mail: liane.hughes@scilifelab.uu.se

†Present address: SciLifeLab Data Centre, Uppsala University Department of Immunology, Genetics and Pathology, Box 815, 751 08, Uppsala, Sweden.

# Increased sperm production linked to competition in the maternal social environment

## Liane Hobson†, Jane L. Hurst and Paula Stockley

Mammalian Behaviour and Evolution Group, Institute of Integrative Biology, University of Liverpool, Leahurst Campus, Chester High Road, Neston CH64 7TE, UK

LH, 0000-0002-4784-5436; JLH, 0000-0002-3728-9624; PS, 0000-0003-1593-5848

Maternal or early life effects may prepare offspring for similar social conditions to those experienced by their mothers. For males, the ability to achieve mating and fertilization success is a key social challenge. Competitive conditions may therefore favour increased body size or ejaculate production in male offspring. We tested this experimentally by comparing reproductive traits of adult male bank voles (*Myodes glareolus*), whose mothers had experienced contrasting encounter regimes with female conspecifics while breeding. We found that daily sperm production rates and epididymis mass were significantly higher when dams had experienced more frequent encounters with female conspecifics. This response to maternal and early life experience was specific to sperm production and storage, with no evidence for effects on male body mass or the size of testes and accessory reproductive glands. Our findings reveal a potentially adaptive effect of maternal and early life experience on the development of sperm production, which is worthy of wider investigation.

## 1. Introduction

The social environment experienced by mothers can have important consequences for the competitive traits of their offspring via maternal and early life effects [1,2]. Such effects can be beneficial by preparing offspring for the particular social environment into which they are born [1,3,4]. Hence if mothers experience social competition, they may produce offspring with competitive phenotypes that are well matched to local conditions. In vertebrates such as placental mammals, there is evidence that such adaptive maternal effects can be triggered by experience during pregnancy. For example, when female red squirrels (*Tamiasciurus hudsonicus*) experience cues of high population density, this leads to increased offspring growth, which is beneficial in competition for limited resources [5]. Similarly, social instability experienced by female guinea pigs

influences the competitive phenotype of their offspring, preparing them for social challenges later in life [4,6].

For male offspring, the ability to win mating opportunities and achieve fertilizations is a key social challenge, for which competitive traits may be subject to influence by maternal effects. Where overt inter-male contests occur, body size can be an important predictor of male mating success [7] and may be influenced by competition in the maternal social environment [5,8]. Although adaptive maternal effects promoting increased body size of sons are likely to be constrained if social competition limits food availability [9,10], they may be selectively favoured if access to sufficient resources is maintained. This might occur for example among territorial species, where competitively successful females are able to defend access to resources required for breeding [11].

Although less well studied than body size in relation to adaptive maternal effects, male ejaculate traits are also an important factor predicting male reproductive success across diverse animal taxa [12]. This is because females of many species typically mate with multiple partners, resulting in competition between the sperm of different males to fertilize their ova [12,13]. Multiple mating and the risk of sperm competition often increase with population density, favouring increased sperm production because this allows males to compete more effectively for fertilizations under competitive conditions [14,15]. Plasticity in sperm production linked to sperm competition risk has previously been linked to cues of population density experienced by males during adolescence [15,16], and it is known that the environment experienced during early development can also affect sperm production in later life [17,18]. Population density has also recently been linked to sperm sex ratios, with increased male density in the maternal and post-maternal environment leading to more Y-chromosome-bearing sperm in male house mice [19]. However, as yet it is unknown if cues of population density experienced by mothers can promote increased sperm production rates in their offspring. Rather, investigations have focused almost exclusively on potential inhibitory influences of maternal experience on offspring fertility, for example, owing to inappropriate maternal nutrition or stress, modelled in laboratory animals [20].

The bank vole, *Myodes glareolus*, is an ideal model to test for potential adaptive maternal and early life effects on the competitive phenotype of male offspring. Adult female bank voles are strongly territorial with respect to other females during the breeding season, defending access to food supplies in support of high energetic costs of reproduction, or sharing the territory with young prior to their dispersal [21,22]. Male home ranges overlap with several females, and there are typically high levels of female multiple mating and resultant sperm competition within natural populations [23,24]. The population density in this species is variable [25,26] and likely to influence levels of sperm competition because females will encounter more males with home ranges overlapping their own at high population density. Social conditions experienced by females during pregnancy, including cues of local population density, are therefore likely to predict the level of reproductive competition that their offspring will experience at reproductive maturity, including the intensity of sperm competition, with potential for adaptive maternal effects to influence male competitive traits.

Here, we investigate if cues of social competition in the environment of breeding female bank voles influence the competitive traits of their male offspring at sexual maturity. Conspecific odours convey important social information regarding both the presence and individual identity of neighbours in rodents, including bank voles [15,16,27]. We therefore predicted that relatively frequent exposure to social odours of other breeding females, combined with short direct encounters, would stimulate maternal or early life effects preparing offspring for a more competitive environment. Specifically, our experiment was designed to test for evidence of adaptive maternal or early life effects on sons' (i) adult body mass, (ii) testis mass, epididymis mass and sperm production, and (iii) seminal vesicle mass, in response to cues of social competition in the maternal environment during pregnancy and lactation. Our results demonstrate evidence of increased sperm production in male offspring whose mothers encountered other females more frequently, which is likely to be adaptive under competitive conditions.

# 2. Methods

## 2.1. Subjects

Female bank voles ($n = 32$) used in this experiment were captive bred first- and second-generation descendants of wild-caught animals trapped in Cheshire, UK. Prior to the experimental treatment, females were weaned at approximately 28 days and housed individually in 45 cm × 28 cm × 13 cm cages (MB1 cage, North Kent Plastics Ltd, UK) for the duration of the experiment, except for mating (§2.2). At the start of the study, they therefore had equivalent prior experience. Females allocated to two treatment groups (low versus high competition, $n = 16$ per group) did not differ in age at the start

of the experiment (mean ± s.e. age: low competition = 253 days ± 15 days, high competition = 250 ± 15 days; Mann–Whitney $U$-test $W = 116.5$, $p = 0.67$). Where females originated from the same litter, siblings were balanced between treatment groups. Male bank voles ($n = 32$) paired with these females were from the same captive colony, consisting of animals that were captive bred ($n = 12$) or wild-caught in Cheshire, UK ($n = 20$), with males of different origin split evenly across the two treatment groups. When not paired with breeding females, males were singly housed in 48 cm × 15 cm × 13 cm cages (M3 cage, North Kent Plastics Ltd, UK). Wild-caught animals were housed under captive conditions for at least four weeks before use. Sixteen litters were produced during the study: eight from high and eight from low-competition treatment groups, although one low-competition litter did not survive to maturity. Male offspring from these litters ($n = 13$ from high- and $n = 16$ from low-competition groups) were individually housed in M3 cages (48 cm × 15 cm × 13 cm, North Kent Plastics Ltd, UK) after weaning at 28 days, prior to analysis of reproductive traits at sexual maturity (§2.3). All cages contained Corn Cob Absorb 10/14 substrate, paper-wool nesting material and a cardboard tube or box for shelter and enrichment. Breeding females were additionally provided with a nest-box (14 cm × 11 cm × 13 cm, Savic, Belgium) when pups were 7 days old to provide additional shelter and minimize disturbance. Food and water were provided ad libitum (LabDiet 5002 Certified Rodent Diet, Purina Mills, USA). Animals were maintained at a temperature of 20 ± 1°C and on a reversed photoperiod (8 : 16 dark : light, with white lights on at 17.00). All contact with animals occurred in the dark period under red lighting and adult animals were handled with Perspex tunnels (approx. 5 cm diameter). Experimental animals were routinely screened for diabetes, which is known to affect bank voles originating from wild populations [28]. At the end of the experiment, female bank voles were returned to the captive colony for further breeding and behavioural studies.

## 2.2. Manipulation of maternal social experience

Both odour cues and direct encounters with other females were used to modify the social experience of female bank voles in this experiment. Olfaction is the primary sensory modality of rodents, and odour cues are likely to contain signals of individual identity [27]. Moreover, scent samples have previously been used successfully to manipulate social experience in rodents, including bank voles [15,16], and bank vole females use odour cues to maintain spacing between territories [29]. Female bank voles exhibit territoriality throughout the breeding season, with females typically occupying individual home ranges or sharing with offspring prior to sexual maturity [21,22]. While odour cues can be used to establish amicable relationships through some of the year, during late pregnancy and lactation females exhibit aggression and scent mark (using urine and probably flank gland secretions) in response to other females even when previously amicable [29]. The experiment was therefore designed to simulate encounters with contrasting numbers of near-neighbours or their odour cues within the females' home range.

Females were randomly assigned to two treatment groups, with the proviso that those from the same parents were balanced between treatment groups. Subjects in the two treatment groups received contrasting social experience for a maximum period of 15 weeks, during which each had the opportunity to mate and rear a litter. During the experiment, females in the high-competition group ($n = 16$) received regular contact with social cues from three different females, whereas those in the low-competition group ($n = 16$) received contact with social cues from only one other female. Females in both groups were also regularly exposed to social cues from the same number of unrelated males, to simulate normal social and sexual behaviour (further details of exposure to social cues are described below). Exposure to social cues other than those presented as part of the experimental treatment group was minimized by placing subjects' cages inside larger high-sided enclosures (w × l × d: 60 cm × 60 cm × 78 cm) within an environmentally controlled room.

The study was conducted in two experimental blocks. Within each block, 16 females were housed within eight enclosures, positioned in a 4 × 2 arrangement to form four rows of two enclosures (the layout of the enclosures is shown in figure 1). Each row contained four individually caged unrelated females, with two cages in each enclosure. Treatment groups were alternated between rows and between experimental blocks to control for spatial positioning. Females were randomly assigned a position within rows, with the proviso that related females were not placed in the same row. As detailed below, throughout the experiment, females from the high-competition group were exposed to social cues from all three other females within the same row, whereas those from the low-competition group were exposed only to the single other female within the same enclosure.

Exposure to odour cues was performed in treatment blocks, each lasting 14 days and commencing weeks 1, 3, 5, 7, 9, 11 and 13, with cages cleaned at the end. Males were introduced in week 7 and

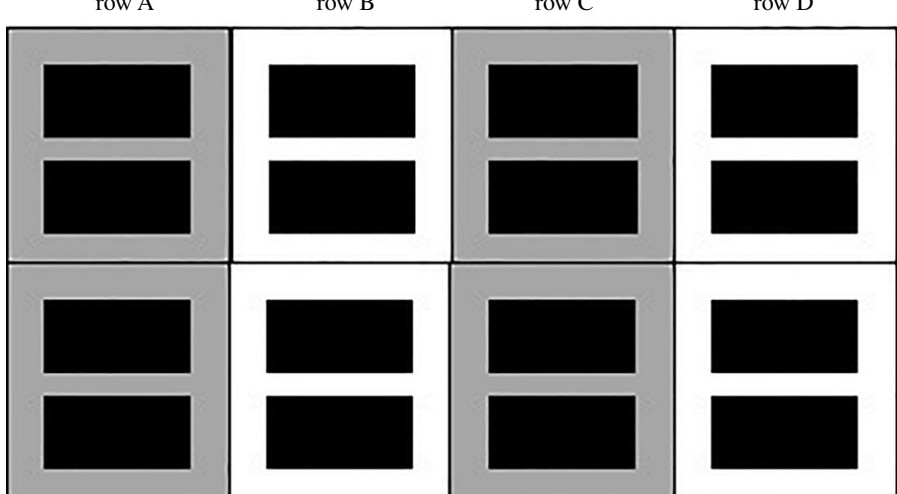

|  | row A | row B | row C | row D |

**Figure 1.** Schematic of enclosure layout. Black rectangles indicate the positions of cages within the high-sided enclosures, represented by large squares. Labels on the top of the image indicate the row name of the high-sided enclosures, and rows with the same colour contained cages of females within the same treatment group.

remained in female cages for one treatment block. Offspring were born in weeks 10–11, and the final treatment block was terminated at offspring weaning. Odour cues were introduced on days 5, 8 and 11 of each treatment block (hereafter 'exposure days'). High-competition females were exposed to scent from a different female on 3 days within each treatment block, with the order of presentation balanced across treatment blocks. Low-competition females were exposed to scent from one female on one exposure day in each treatment block, with the exposure day balanced between blocks, and were provided with their own scent on the other two exposure days. Odour cues consisted of 25 g of soiled cage substrate from the rear section of donor home cages, introduced into the front section of recipient home cages. Soiled substrate was left in the recipient's cage until the cage was cleaned at the end of the treatment block. To prevent the transfer of male scent during the mating period (weeks 7–9), we removed the required number of scent samples from each donor cage at the end of week 6. Samples were stored at −20°C and defrosted before use.

Females were also exposed to a scent sample from an unrelated male on day 1 of treatment blocks commencing weeks 1, 3, 5, 9 and 11, to simulate natural exposure to male social cues. As exposure to conspecific odour cues continued after pups were born, we cannot rule out the possibility that pups in the nest were also directly exposed to odour cues. Hence, we refer to the manipulation of levels of competition in the maternal environment, rather than to maternal competition *per se* (see discussion).

Controlled encounters were also arranged between females that received odour cues from one another. Encounters lasted for 30 min and occurred on day 1 of treatment blocks commencing weeks 1, 3, 5 and 7. Encounters took place in a neutral enclosure that was subdivided into four compartments of equal area (0.36 m$^2$), each linked to the other three by circular holes covered with mesh (approx. 5 cm diameter, 1 cm from enclosure floor). This enclosure is illustrated in figure 2. This allowed female bank voles to contact each other without the risk of harm. The order in which females were transferred to the enclosure and the compartment into which they were placed was balanced across encounters.

## 2.3. Sons' reproductive traits

Male offspring were housed individually after weaning (§2.1). Male bank voles reach sexual maturity at approximately 8 weeks of age [30]. Male offspring were killed humanely at age 11 weeks and measurements of body mass, and the mass of the epididymides, testes and seminal vesicles were taken. These measurements were made blind to treatment. Three males (one from the low-competition group, and two from the high-competition group) were excluded from further analysis after testing positive for diabetes at age 10–11 weeks (§2.1). These males were also noted to have smaller epididymides than other males at dissection and were significant outliers with respect to the mass of epididymides.

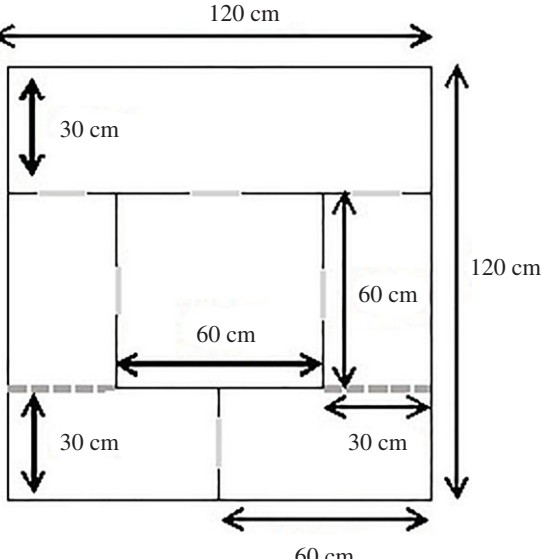

**Figure 2.** Schematic of the encounter enclosure. Solid enclosure walls (78 cm high) are depicted using black lines. All outer walls are 1.5 cm thick, and inner walls are 0.3 cm thick. Dashed grey lines represent 'half walls', where there was only a solid wall over the upper 50 cm of the enclosure. Adult females could move freely under half walls. Solid grey lines indicate the positions of circular 'windows'. These 'windows' were filled with a circular piece of wire mesh, which was secured with four cable ties. Each window was 10 cm in diameter and located 1 cm from the enclosure floor. Measures of the lengths of each part of the enclosure are provided on black arrows.

Daily sperm production rate was approximated using a method adapted from Seung *et al.* [31] and described in Ramm & Stockley [15]. The right testicle of each male was stored at −20°C immediately after dissection. The tunica albuginea was removed after testes had thawed for 1 min. The remaining tissue was then weighed and homogenized in 10 ml of dimethyl-sulphoxide/saline solution using a Ystral X10/20 homogenizer with a 10 T shaft (Yastral, Germany). Homogenization occurred in two stages, each lasting 60 s. Spermatids were stained with Trypan blue, and spermatid heads were counted under 40 x magnification using a Neubauer haemocytometer (Neubauer, USA). Spermatid head counts were performed blind to treatment.

## 2.4. Statistical analyses

Linear mixed-effects models were performed using the lme4 package in R [32] to test for evidence that the maternal social environment influences body size or ejaculate investment of male offspring. In each model, the experimental block and maternal identity were included as random effects, and the treatment group was included as a fixed effect. Male body mass was included as a fixed effect in analyses assessing the size of reproductive organs, and testis mass was included as a fixed effect in models assessing daily sperm production rate.

To test for effects of maternal treatment group on sons' reproductive traits, we used likelihood ratio tests to compare a 'full' model (i.e. one containing all random and fixed effects) to a 'test' model (i.e. one that contained all random and fixed effects except treatment). Where appropriate, values were log-transformed to improve normality.

Mann–Whitney $U$-tests were used to confirm that there was no difference in the ages of the mothers in each group prior to testing, to test for a difference in the number of sons per litter between treatment groups and to examine potential differences in litter sex ratio between treatment groups. Binomial tests were used to further explore potential differences in litter sex ratio by testing whether the ratio differed from equality.

## 3. Results

We found evidence that the maternal social environment influences the sperm production of male offspring. Males born in the high-competition group had significantly larger epididymides after

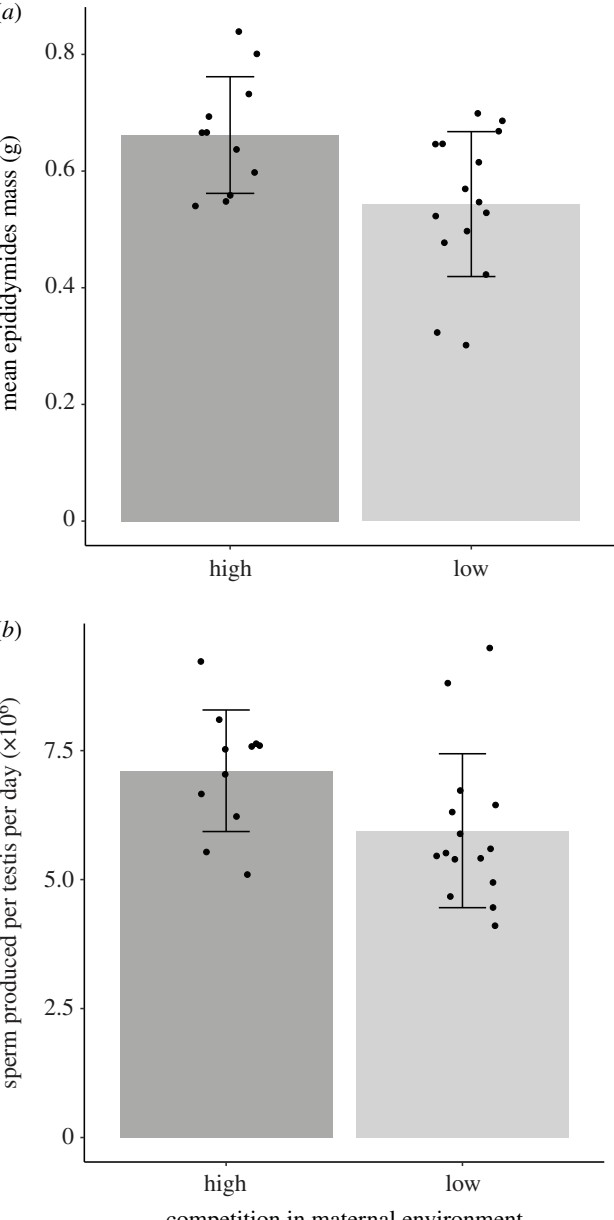

**Figure 3.** Comparison of sperm production traits for sexually mature male bank voles born to mothers in high and low-competition treatment groups. Male offspring of females in the high-competition group had significantly greater (*a*) epididymides masses ($\chi^2_1 =$ 6.07, $p = 0.01$) and (*b*) daily sperm production rates ($\chi^2_1 = 4.70$, $p = 0.03$).

controlling for body mass (combined epididymis mass mean ± s.e.: high competition = 0.66 ± 0.03 g, low competition = 0.54 ± 0.03 g; $\chi^2_1 = 6.07$, $p = 0.01$; figure 3*a*; electronic supplementary material, table S1) and significantly higher daily sperm production rates compared to those born in the low-competition group (mean ± s.e. sperm d$^{-1}$; high competition: $7.11 \times 10^6 \pm 3.6 \times 10^5$; low competition: $5.95 \times 10^6 \pm 3.9 \times 10^5$; $\chi^2_1 = 4.70$, $p = 0.03$; figure 3*b*; electronic supplementary material, table S2) after controlling for testes mass.

Body mass of male offspring did not differ between treatment groups (mean ± s.e. adult body mass: high competition = 20.29 ± 0.39 g, low competition = 19.74 ± 0.34 g; $\chi^2_1 = 0.045$, $p = 0.83$; electronic supplementary material, table S3). Similarly, with control for body mass, there was no evidence for a treatment effect on the size of testes (mean combined testes mass ± s.e.: high competition = 0.52 ± 0.01 g; low competition = 0.52 ± 0.01 g, $\chi^2_1 = 0.52$, $p = 0.47$; electronic supplementary material, table S4) or seminal vesicles (mean combined seminal vesicles mass ± s.e.: high competition = 0.16 ± 0.01 g, low competition = 0.14 ± 0.01 g, $\chi^2_1 = 0.64$, $p = 0.42$; electronic supplementary material, table S5).

Finally, since exposure to androgens *in utero* from male siblings can influence sperm production traits [17], we tested if the number of sons per litter or litter sex ratio differed according to treatment group. However, there was no difference between treatment groups in the number of sons per litter (mean ± s.e. number of male offspring per litter: high competition = 1.62 ± 0.32; low competition = 2.29 ± 0.29, Mann–Whitney *U*-test $W = 16.5$, $p = 0.17$). While there was a significant difference in sex ratio between treatment groups ($W = 8$, $p = 0.022$), sex ratio was found to be female-biased in litters of high-competition females (mean ± s.e. litter sex ratio: high competition = 0.354 ± 0.07; low competition = 0.764 ± 0.12). However, binomial tests showed that litter sex ratio did not differ significantly from equality (i.e. 0.5) in either treatment group (high competition: $p = 0.176$; low competition; $p = 0.093$).

## 4. Discussion

Our experiment provides evidence that cues of competition in the maternal social environment can lead to increased sperm production by sons. Female bank voles experiencing social cues associated with high-competition produced sons with significantly higher rates of sperm production at sexual maturity and significantly heavier epididymides. By contrast, we found no significant difference in the adult body mass of sons produced under contrasting social conditions, or in the mass of their testes and seminal vesicles at sexual maturity.

Our findings expand current evidence of social influences on ejaculate production in vertebrate animals to include early life effects. Previous studies have focused primarily on how social conditions experienced by males during sexual development or adulthood influence investment in ejaculates (e.g. [15,16,33,34]). For example, male house mice experiencing social cues associated with high competition (number of males encountered) during sexual development have higher sperm production rates and a different composition of seminal fluid compared to males experiencing social cues of low competition [15,33]. Similarly, male bank voles develop larger seminal vesicles when exposed to social cues associated with high social competition during sexual development [16]. In each case, the reported effects are likely to be adaptive because increased sperm production rates and/or larger seminal vesicles promote male success under sperm competition, the risk of which is increased at high population density as females of these species mate with multiple males [35]. Increased investment in ejaculate production may also facilitate an increased rate of multiple mating by males, although previous studies suggest a stronger response to cues of competition than likely mating rate [33].

Tests of how maternal social experience affects the reproductive traits of sons are more limited, with emphasis on how maternal stress might negatively impact offspring traits such as the timing of sexual maturation, rather than on beneficial traits under sperm competition. For example, Marchlewska-Koj *et al.* [36] exposed female bank voles to short frequent encounters with other females as a stressor during pregnancy and found evidence of delayed sexual maturation in male offspring. Similarly, using social instability as a stressor during pregnancy in wild cavies, Siegeler *et al.* [37] found that sons whose mothers had experienced an unstable social environment showed a delayed increase in testosterone levels around adolescence compared to controls. However, no difference was found in adult testes or epididymides masses according to maternal social experience, or in a range of fertility parameters, including sperm motility. Other studies have used crowding to induce social stress during pregnancy, with resultant reductions in fertility of male offspring via their sexual behaviour (e.g. [38]). By contrast to these previous studies, to our knowledge, the findings of the present study provide the first evidence of increased sperm production, with potentially adaptive benefits, linked to maternal social experience. Increased sperm production is likely to be beneficial to male fitness under high population density, both by facilitating multiple mating and by promoting fertilization success under an elevated risk of sperm competition [35].

Although we did not investigate potential mechanisms underlying increased sperm production by male offspring, previous studies have linked maternal effects in vertebrate animals to maternal hormone expression during pre- and postnatal development [39,40]. For example, elevated maternal glucocorticoids, as a response to stress, have well-established effects on offspring development [41,42]. In natural populations, these may include adaptive effects on offspring growth rate [5,8]. Our own study was designed to simulate naturalistic social conditions experimentally, with relatively infrequent direct encounters with conspecifics, because female bank voles are sensitive to social odour cues and typically avoid direct contact with one another during the breeding season [21]. Hence if subjects perceived the high-competition environment in our experiment as relatively stressful, it seems likely that any fluctuation in their glucocorticoid levels would have fallen within the normal range associated with typical social

experience in the wild. Nonetheless, despite the potential for some variation in the levels of stress experienced by subjects between our experimental treatment groups, we found no evidence for differences in the eventual body size of their offspring. Maternal nutrition can also have effects on offspring reproductive traits, as well as growth [20], but this can be ruled out as a factor explaining maternal effects observed in the current experiment, because all females had ad libitum access to a balanced diet.

Exposure to androgens during early development can have long-term effects on reproductive traits, including sperm production [20]. Hence if bank vole mothers respond to high levels of social competition with elevated androgen levels during pregnancy, this could influence the sexual development of their sons. Among avian species, aggressive competition with conspecifics has been shown to result in increased maternal allocation of testosterone to egg yolk, which in turn can lead to increased competitive traits in offspring [43,44]. In laboratory and domesticated mammals, artificially increased prenatal exposure to excess androgens can inhibit testes size and sperm production in later life [45,46]. However, in those studies, animals were exposed to extremely high levels of androgens *in utero*. Other studies, involving more subtle variation in androgen exposure have shown different outcomes. For example, it is well known that the development of male rodents is influenced by their intra-uterine position; that is, male foetuses positioned between two brothers ultimately develop larger testes as a result of higher prenatal androgen exposure [47,48], and there is evidence that males with more male siblings *in utero* also have increased fertility [17]. While we cannot exclude the possibility of passive exposure to androgens from siblings *in utero* as an explanation for our results, it seems unlikely. Males in our high-competition treatment group were not exposed to more male siblings *in utero*. Rather, our data indicates that offspring sex ratio was, if anything, more female-biased in the high-competition group. However, the number of litters tested here was not sufficiently large to permit robust conclusions about sex ratio variation, and litter sex ratio did not differ significantly from equality. Our findings thus instead suggest the possibility that intraspecific variation in sperm production might be sensitive to natural variation in maternal androgen levels, as a response to competitive conditions.

Although perhaps a less likely explanation, offspring development might also be influenced by direct exposure to odour cues, either passively transferred by their mother, or during pre-weaning forays from the nest. Whether this would be classed as a maternal effect is debatable [2]. However, at least under natural conditions, a mother's behaviour and choice of nest site should ultimately determine the extent to which her offspring are exposed to social odours prior to weaning. If direct exposure to female odour cues during early life affects sperm production rates, this could be an adaptive response to an elevated risk of sperm competition, as a generalized response to cues of high population density, and/or an adaptive response specifically to social cues of local breeding females, to facilitate an increased mating rate.

Finally, we note that the response to maternal and early life experience reported here was specific to sperm production and storage in male offspring, with no evidence for effects on body mass or the size of testes and accessory reproductive glands. Although it is well known that sperm production rates can vary independently of testis size [15,49], it is less clear why no differences were found in the size of the seminal vesicles according to early life experience, because larger accessory reproductive glands should be favoured both to promote sperm competition success and an increased mating rate [16,33]. Further explanation requires a more detailed investigation of the mechanisms involved, both with respect to how plasticity in reproductive traits is mediated at different stages of development and the adaptive consequences of such plasticity in a competitive context.

In conclusion, our finding that the maternal social environment can influence adult sperm production offers a new perspective for understanding the plasticity of ejaculate traits in vertebrate animals and is worthy of a wider investigation.

Ethics. All animal care and experimental protocols were in accordance with the University of Liverpool Animal Welfare Committee requirements, EU Directive 2010/63/EU and the UK Home Office code of practice for the housing and care of animals bred, supplied or used for scientific purposes. No specific licences were required to carry out the work, which adheres to the Association for the Study of Animal Behaviour/Animal Behaviour Society Guidelines for the Use of Animals in Research.

Data accessibility. The full dataset supporting this article is available on Dryad: https://dx.doi.org/10.5061/dryad. 547d7wm53 [50].

Authors' contributions. L.H. participated in the design of the study, collected and analysed data and drafted the manuscript; J.L.H. participated in the design of the study and critically revised the manuscript; P.S. conceived of the study, participated in the design of the study, helped to draft the manuscript and secured funding. All authors gave final approval for publication and agree to be held accountable for the work performed therein.

Competing interests. We declare we have no competing interests.

Funding. The study was funded by a research grant (grant no. NE/1013008/1) and studentship from the Natural Environment Research Council UK.

Acknowledgements. We thank Olivia Antony and all members of the Mammalian Behaviour and Evolution Group of the University of Liverpool, including Rachel Spencer, John Waters and Joshua Beeston for animal care, and Amanda Davidson and Rick Humphries for technical support. We are also grateful to Jeff Lemaître and anonymous reviewers for their extremely helpful comments.

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
