## [Reviewer comments · Royal Society Open Science]

Review History

RSOS-201171.R0 (Original submission)

Review form: Reviewer 1

Is the manuscript scientifically sound in its present form?

Yes

Are the interpretations and conclusions justified by the results?

Yes

Is the language acceptable?

Yes

Do you have any ethical concerns with this paper?

No

Have you any concerns about statistical analyses in this paper?

Yes

Recommendation?

Accept with minor revision (please list in comments)

Comments to the Author(s)

In the manuscript 'Increased sperm production linked to competition in the maternal social environment' Hobson and colleagues report the intriguing observation that sons born to mothers who experience more social encounters with other females during gestation and lactation have larger epididymides and higher sperm production rates than sons born to mothers experiencing very few encounters.

The manuscript is generally well written, the experiment was well designed and executed (great that the measurements were done blind!), and the limitations of the study are discussed. The idea is original, and the observed effect very interesting. Unfortunately, the number of litters from which measurements were taken is rather small, even for a vertebrate study, mostly due to only half of the females subjected to the experimental treatments giving birth. This raises some doubts about the robustness of the findings.

This re-submitted MS has already been reviewed by two other reviewers, and has been transferred from Proceedings B. Those two reviewers' comments have IMO been sensibly addressed.

The term 'maternal social environment' in my view implies that only the mother's environment was manipulated, but the authors do not hide the fact that early post-natal environment was also changed according to treatment, so for the sake of brevity I think maternal social environment is adequate.

Personally, I too would prefer figures that include raw data. Data being available on dryad is IMO not a good argument for not producing a figure that is (probably) more useful for readers. [NB: the data pertaining to sex ratio is not detailed enough to reproduce the analyses presented, as only the number of sons is provided, without number of daughters or litter size]

I have two issues that caused a bit of confusion for me, though they are quite minor:

The first is that the authors state in their response to reviewers that they "1) added an extra analysis to confirm that there is no difference in sex ratio within litters according to treatment groups", but they also report that "[...] there was a significant difference in sex ratio between treatment groups ($W = 8$, $P = 0.022$)" [on a related note, why not also use a binomial test here?]. They then go on to say that neither groups differed from a 50:50 sex ratio. However, given that litters were quite small (which might be normal for bank voles), and that there were only 8 and 7 litters in the high and low treatment, respectively (according to "Number_of_males_per_litter_data.xlsx"; the main text mentions 16 litters!?), the power to detect a deviation from equality is rather low. Given this, IMO the test against 50:50 should not be used as an argument. However, I agree with the argument that high treatment showing a female bias is counter to what would be predicted if sex ratio affected the main results.

Second, the statement that "*These males [testing positive for diabetes] were also noted to have smaller epididymides than other males at dissection, and were significant outliers with respect to the mass of epididymides and daily rate of sperm production*" is not fully backed up by the deposited data (see uploaded plots). The diabetic males do appear to be outliers for epididymides mass relative to body mass [NB: the significance of the difference between treatments relies on removing these males], but their sperm production rate relative to testis mass appear to lie within the normal variation. Again, while removing diabetic individuals seems sensible, the statement about sperm is misleading (unless I misunderstood what is meant).

Finally, perhaps the authors could add p-values to their tables in the ESM? For example, loading the *lmerTest* package before running the *lme4* models will give p values in the output, with degrees of freedom estimated using an approximation. While not identical to the LRT approach reported by the authors, qualitatively this should give similar results. Alternatively, the authors could highlight significant effects in bold.

Review form: Reviewer 2 (Jean-François Lemaitre)

Is the manuscript scientifically sound in its present form?

Yes

Are the interpretations and conclusions justified by the results?

Yes

Is the language acceptable?

Yes

Do you have any ethical concerns with this paper?

No

Have you any concerns about statistical analyses in this paper?

No

Recommendation?

Accept with minor revision (please list in comments)

Comments to the Author(s)

Dear authors,

Please find below my comments on your paper 'Increased sperm production linked to competition in the maternal social environment'. This paper has already been reviewed by two reviewers and the comments raised by these two reviewers have been addressed very carefully. Overall, I have found this article very interesting and I think it will open the door to future studies investigating how maternal effects can shape son's reproductive biology.

I just have one comment regarding the analyses. As rightly stated by the authors, androgens exposure in utero can influence male reproductive physiology. Therefore, when testing for the effect of the treatment group on the male reproductive traits, the litter sex ratio should be added as a potential covariate in the full model. In fact, the male allocation to each of reproductive trait investigated in this study could depend of an interaction between the treatment group and the litter-sex ratio.

Kind regards,
Jean-François Lemaître

Decision letter (RSOS-201171.R0)

Dear Dr Hobson

On behalf of the Editors, we are pleased to inform you that your Manuscript RSOS-201171 "Increased sperm production linked to competition in the maternal social environment" has been accepted for publication in Royal Society Open Science subject to minor revision in accordance with the referees' reports. Please find the referees' comments along with any feedback from the Editors below my signature.

Please submit your revised manuscript and required files (see below) no later than 7 days from today's (ie 09-Nov-2020) date. Note: the ScholarOne system will 'lock' if submission of the revision is attempted 7 or more days after the deadline. If you do not think you will be able to meet this deadline please contact the editorial office immediately.

on behalf of the Associate Editor and Professor Kevin Padian (Subject Editor)
openscience@royalsociety.org

Associate Editor Comments to Author:

Thank you for submitting this manuscript to the journal after transfer from Proceedings B - the reviewers now recommend minor revisions. We'll look forward to receiving the revised paper in the near future.

Reviewer comments to Author:

Reviewer: 1
Comments to the Author(s)

In the manuscript 'Increased sperm production linked to competition in the maternal social environment' Hobson and colleagues report the intriguing observation that sons born to mothers who experience more social encounters with other females during gestation and lactation have larger epididymides and higher sperm production rates than sons born to mothers experiencing very few encounters.

The manuscript is generally well written, the experiment was well designed and executed (great that the measurements were done blind!), and the limitations of the study are discussed. The idea is original, and the observed effect very interesting. Unfortunately, the number of litters from which measurements were taken is rather small, even for a vertebrate study, mostly due to only half of the females subjected to the experimental treatments giving birth. This raises some doubts about the robustness of the findings.

This re-submitted MS has already been reviewed by two other reviewers, and has been transferred from Proceedings B. Those two reviewers' comments have IMO been sensibly addressed.

The term 'maternal social environment' in my view implies that only the mother's environment was manipulated, but the authors do not hide the fact that early post-natal environment was also changed according to treatment, so for the sake of brevity I think maternal social environment is adequate.

Personally, I too would prefer figures that include raw data. Data being available on dryad is IMO not a good argument for not producing a figure that is (probably) more useful for readers. [NB: the data pertaining to sex ratio is not detailed enough to reproduce the analyses presented, as only the number of sons is provided, without number of daughters or litter size]

I have two issues that caused a bit of confusion for me, though they are quite minor:

The first is that the authors state in their response to reviewers that they "**1**) added an extra analysis to confirm that there is no difference in sex ratio within litters according to treatment groups", but they also report that "[...] there was a significant difference in sex ratio between treatment groups ($W = 8$, $P = 0.022$)" [on a related note, why not also use a binomial test here?]. They then go on to say that neither groups differed from a 50:50 sex ratio. However, given that litters were quite small (which might be normal for bank voles), and that there were only 8 and 7 litters in the high and low treatment, respectively (according to "Number_of_males_per_litter_data.xlsx"; the main text mentions 16 litters!), the power to detect a deviation from equality is rather low. Given this, IMO the test against 50:50 should not be used as an argument. However, I agree with the argument that high treatment showing a female bias is counter to what would be predicted if sex ratio affected the main results.

Second, the statement that "These males [testing positive for diabetes] were also noted to have smaller epididymides than other males at dissection, and were significant outliers with respect to the mass of epididymides and daily rate of sperm production" is not fully backed up by the deposited data (see uploaded plots). The diabetic males do appear to be outliers for epididymides mass relative to body mass [NB: the significance of the difference between treatments relies on removing these males], but their sperm production rate relative to testis mass appear to lie within the normal variation. Again, while removing diabetic individuals seems sensible, the statement about sperm is misleading (unless I misunderstood what is meant).

Finally, perhaps the authors could add p-values to their tables in the ESM? For example, loading the `lmerTest` package before running the `lme4` models will give p values in the output, with degrees of freedom estimated using an approximation. While not identical to the LRT approach reported by the authors, qualitatively this should give similar results. Alternatively, the authors could highlight significant effects in bold.

Reviewer: 2

Comments to the Author(s)

Dear authors,

Please find below my comments on your paper 'Increased sperm production linked to competition in the maternal social environment'. This paper has already been reviewed by two reviewers and the comments raised by these two reviewers have been addressed very carefully. Overall, I have found this article very interesting and I think it will open the door to future studies investigating how maternal effects can shape son's reproductive biology.

I just have one comment regarding the analyses. As rightly stated by the authors, androgens exposure in utero can influence male reproductive physiology. Therefore, when testing for the effect of the treatment group on the male reproductive traits, the litter sex ratio should be added as a potential covariate in the full model. In fact, the male allocation to each of reproductive trait

investigated in this study could depend of an interaction between the treatment group and the litter-sex ratio.

Kind regards,
Jean-François Lemaître

===PREPARING YOUR MANUSCRIPT===

===PREPARING YOUR REVISION IN SCHOLARONE===

- 1) One version identifying all the changes that have been made (for instance, in coloured highlight, in bold text, or tracked changes);
 - 2) A 'clean' version of the new manuscript that incorporates the changes made, but does not highlight them.
 - An individual file of each figure (EPS or print-quality PDF preferred [either format should be produced directly from original creation package], or original software format).
 - An editable file of each table (.doc, .docx, .xls, .xlsx, or .csv).
 - An editable file of all figure and table captions.
- Note: you may upload the figure, table, and caption files in a single Zip folder.
- Any electronic supplementary material (ESM).
 - If you are requesting a discretionary waiver for the article processing charge, the waiver form must be included at this step.
 - If you are providing image files for potential cover images, please upload these at this step, and inform the editorial office you have done so. You must hold the copyright to any image provided.
 - A copy of your point-by-point response to referees and Editors. This will expedite the preparation of your proof.

- Ensure that your data access statement meets the requirements at <https://royalsociety.org/journals/authors/author-guidelines/#data>. You should ensure that you cite the dataset in your reference list. If you have deposited data etc in the Dryad repository, please only include the 'For publication' link at this stage. You should remove the 'For review' link.
- If you are requesting an article processing charge waiver, you must select the relevant waiver option (if requesting a discretionary waiver, the form should have been uploaded at Step 3 'File upload' above).
- If you have uploaded ESM files, please ensure you follow the guidance at <https://royalsociety.org/journals/authors/author-guidelines/#supplementary-material> to include a suitable title and informative caption. An example of appropriate titling and captioning may be found at https://figshare.com/articles/Table_S2_from_Is_there_a_trade-off_between_peak_performance_and_performance_breadth_across_temperatures_for_aerobic_scope_in_teleost_fishes_/3843624.

Author's Response to Decision Letter for (RSOS-201171.R0)

See Appendix A.

Decision letter (RSOS-201171.R1)

Dear Dr Hobson,

It is a pleasure to accept your manuscript entitled "Increased sperm production linked to competition in the maternal social environment" in its current form for publication in Royal Society Open Science.

on behalf of Professor Kevin Padian (Subject Editor)
openscience@royalsociety.org

Appendix A

Reviewer comments to Author:

Reviewer: 1

Comments to the Author(s)

In the manuscript 'Increased sperm production linked to competition in the maternal social environment' Hobson and colleagues report the intriguing observation that sons born to mothers who experience more social encounters with other females during gestation and lactation have larger epididymides and higher sperm production rates than sons born to mothers experiencing very few encounters.

The manuscript is generally well written, the experiment was well designed and executed (great that the measurements were done blind!), and the limitations of the study are discussed. The idea is original, and the observed effect very interesting. Unfortunately, the number of litters from which measurements were taken is rather small, even for a vertebrate study, mostly due to only half of the females subjected to the experimental treatments giving birth. This raises some doubts about the robustness of the findings.

Whilst our study was theoretically well powered at the design stage, the animals did not breed as well as laboratory rodents typically do. Because of the resultant low number of litters, and the potential consequences of this on the robustness of results, we have been careful to ensure that the tone of the paper is such that we do not over-state our conclusions. Nonetheless, as both reviewers agree, our study has provided some very interesting findings.

This re-submitted MS has already been reviewed by two other reviewers, and has been transferred from Proceedings B. Those two reviewers' comments have IMO been sensibly addressed.

The term 'maternal social environment' in my view implies that only the mother's environment was manipulated, but the authors do not hide the fact that early post-natal environment was also changed according to treatment, so for the sake of brevity I think maternal social environment is adequate.

We are grateful to the reviewer for this assessment and their thoughts on the term we chose to use.

Personally, I too would prefer figures that include raw data. Data being available on dryad is IMO not a good argument for not producing a figure that is (probably) more useful for readers. [NB: the data pertaining to sex ratio is not detailed enough to reproduce the analyses presented, as only the number of sons is provided, without number of daughters or litter size]

We are of course happy to provide data on sex ratio, and this data has now been submitted to Dryad alongside our other data files. Figure 3 has been modified so that raw data are shown on the graph.

I have two issues that caused a bit of confusion for me, though they are quite minor:

The first is that the authors state in their response to reviewers that they "*1) added an extra analysis to confirm that there is no difference in sex ratio within litters according to treatment groups*", but they also report that "*[...] there was a significant difference in sex ratio between treatment groups ($W = 8, P = 0.022$)*" [on a related note, why not also use a binomial test here?]. They then go on to say that neither groups differed from a 50:50 sex ratio. However, given that litters were quite small (which might be normal for bank voles), and that there were only 8 and 7 litters in the high and low treatment, respectively (according to "Number_of_males_per_litter_data.xlsx"; the main text

mentions 16 litters!?), the power to detect a deviation from equality is rather low. Given this, IMO the test against 50:50 should not be used as an argument. However, I agree with the argument that high treatment showing a female bias is counter to what would be predicted if sex ratio affected the main results.

We completed two different types of analysis for sex ratio, as is recommended for sex ratio analysis. As stated in the manuscript (lines 258-266), whilst we found a significant difference when using the Mann-Whitney U test, the binomial test that we included showed no significant deviation from a 50:50 sex ratio. This is why we concluded that the treatment applied did not have a significant effect on litter sex ratio.

We agree with the reviewer that our power to detect a difference is low. However, as the reviewer states, the findings related to sex ratio do not indicate that a male-biased ratio would have become evident with the addition of more data. Due to the low numbers of litters tested, we have ensured that the tone does not over-state the results. However, we have now further highlighted this point (346-349) in the context of the sex ratio data.

The reason that only 15 litters are included in the data, as noted by the reviewer, is that one litter was lost before measurements could be taken. This has been clarified (line 113).

Second, the statement that *“These males [testing positive for diabetes] were also noted to have smaller epididymides than other males at dissection, and were significant outliers with respect to the mass of epididymides and daily rate of sperm production”* is not fully backed up by the deposited data (see uploaded plots). The diabetic males do appear to be outliers for epididymides mass relative to body mass [NB: the significance of the difference between treatments relies on removing these males], but their sperm production rate relative to testis mass appear to lie within the normal variation. Again, while removing diabetic individuals seems sensible, the statement about sperm is misleading (unless I misunderstood what is meant).

We thank the reviewer for this insight and have now modified the text (line 208=9) to reflect that we only found direct evidence for an effect of diabetes on epididymides mass. We agree with the reviewer that it remains sensible to have excluded these males from the analysis.

Finally, perhaps the authors could add p-values to their tables in the ESM? For example, loading the *lmerTest* package before running the *lme4* models will give p values in the output, with degrees of freedom estimated using an approximation. While not identical to the LRT approach reported by the authors, qualitatively this should give similar results. Alternatively, the authors could highlight significant effects in bold.

We appreciate the reviewer’s thoughts on the potential inclusion of P values in the ESM. However, we do not feel that it would be appropriate to include P-values generated using a different method than the approach we used in our study. The P-values stated in the main text refer to the difference between the models when compared (which specifically examines the effect of treatment). It could therefore be misleading to highlight the values for the effect of treatment in the ESM, as the effect of the other factors included in the full model are not expressly considered in the method that we used. As a result, we have decided that it would not be appropriate to highlight these values.

Reviewer: 2

Comments to the Author(s)

Dear authors,

Please find below my comments on your paper 'Increased sperm production linked to competition in the maternal social environment'. This paper has already been reviewed by two reviewers and the comments raised by these two reviewers have been addressed very carefully. Overall, I have found this article very interesting and I think it will open the door to future studies investigating how maternal effects can shape son's reproductive biology.

I just have one comment regarding the analyses. As rightly stated by the authors, androgens exposure in utero can influence male reproductive physiology. Therefore, when testing for the effect of the treatment group on the male reproductive traits, the litter sex ratio should be added as a potential covariate in the full model. In fact, the male allocation to each of reproductive trait investigated in this study could depend of an interaction between the treatment group and the litter-sex ratio.

We thank the reviewer for his comment. We have considered this point at length, and on balance have decided not to include sex ratio as a factor in our models. As highlighted by Reviewer 1, our sample size is quite low and more data would be required for our models to support the inclusion of additional factors. Moreover, sex ratio is likely to be a relatively crude proxy for the effect of individual position in utero (i.e. next to zero, one or two male siblings – which is the real factor of interest in this context), hence a much larger sample would likely be required to make this a useful covariate. Sex ratio was not a factor in our original hypothesis or experimental design, and as an added complication, our models include maternal identity, which co-varies with sex ratio. So rather than include sex ratio as a covariate in the models, we have instead tested if it is likely to be a potential cofound in our experiment using separate Man-Whitney U and binomial tests (258-266)). Here, all evidence indicates that, if anything, sex ratio is impacted in the opposite way than may have been expected and is thus unlikely to explain the patterns reported.